# Are Reallocations between Sedentary Behaviour and Physical Activity Associated with Better Sleep in Adults Aged 55+ Years? An Isotemporal Substitution Analysis

**DOI:** 10.3390/ijerph17249579

**Published:** 2020-12-21

**Authors:** Julie Vanderlinden, Gregory J. H. Biddle, Filip Boen, Jannique G. Z. van Uffelen

**Affiliations:** 1Physical Activity, Sports and Health Research Group, Department of Movement Sciences, KU Leuven, University of Leuven, 3000 Leuven, Belgium; filip.boen@kuleuven.be (F.B.); jannique.vanuffelen@kuleuven.be (J.G.Z.v.U.); 2Department of Health Care, Odisee University College, 1000 Brussels, Belgium; 3School of Sport, Exercise and Health Sciences, Loughborough University, Loughborough LE11 3TU, UK; G.J.Biddle@lboro.ac.uk

**Keywords:** sleep, physical activity, sedentary behaviour, isotemporal substitution analysis, older adults, public health

## Abstract

Physical activity has been proposed as an effective alternative treatment option for the increasing occurrence of sleep problems in older adults. Although higher physical activity levels are associated with better sleep, the association between specific physical activity intensities and sedentary behaviour (SB) with sleep remains unclear. This study examines the associations of statistically modelled time reallocations between sedentary time and different physical activity intensities with sleep outcomes using isotemporal substitution analysis. Device-measured physical activity data and both objective and subjective sleep data were collected from 439 adults aged 55+ years. Replacing 30 min of SB with moderate to vigorous intensity physical activity (MVPA) was significantly associated with an increased number of awakenings. Moreover, a reallocation of 30 min between light physical activity (LPA) and MVPA was significantly associated with increased sleep efficiency. Furthermore, reallocating 30 min of SB to LPA showed a significant association with decreased sleep efficiency. There were no significant associations of time reallocations for wake time after sleep onset, length of awakenings, and sleep quality. These results improve our understanding of the interrelationships between different intensities of movement behaviours and several aspects of sleep in older adults.

## 1. Introduction

Ageing is associated with an increased prevalence of sleep problems [1,2]. Approximately 50% of older adults suffer from these age-related changes in sleep, including (1) spending more time in bed but less time asleep, (2) more disrupted and less deep sleep, and (3) more frequent early risings [3,4,5]. These changes in sleep are problematic, as they cause fatigue, daytime sleepiness, and excess napping during the day, which in turn can influence the sleep at night [6,7]. In addition, they can negatively affect physical functioning and quality of life, as well as cognitive function and mental health [1,4,6,8,9]. Most sleep problems in older adults are currently treated with medication [10]. However, these medications can cause side effects, such as blurred vision, dizziness, sedation, anxiety, and fatigue [11,12,13]. Moreover, these medications are not always effective or safe in the long-term [11,12,13]. From a public health perspective, non-pharmaceutical treatment alternatives, such as cognitive behavioural therapy, sleep hygiene advice, relaxation exercises, and physical activity [9], are therefore the preferred choice for preventing or managing these age-related changes in sleep [13].

Physical activity (PA) has been defined by Caspersen et al. (1985) as any bodily movement produced by skeletal muscles that results in energy expenditure [14]. These movements can be classified as light, moderate, or vigorous intensity, depending on the amount of energy expended by an individual during the activity [14,15,16]. PA has been associated with both improved objective sleep outcomes, including wake time after sleep onset (WASO), as well as with improved subjective sleep outcomes, such as sleep quality, sleep latency, and sleep disturbances in older adults [9,17,18,19]. More specifically, regular PA has been found to promote relaxation and energy expenditure in ways that are beneficial for sleep [20,21]. Based on these findings, PA might thus constitute an effective non-pharmaceutical option for improving sleep in older adults [21].

Although several studies reported associations between PA and sleep, there is uncertainty about the specific PA intensity and its association with sleep. Light intensity physical activity (LPA) might be preferable because of better compliance, lower risk of injuries, and long-term sustainability [9,22,23]. Indeed, recent studies reported that LPA is associated with better sleep outcomes, such as sleep quality, sleep latency, sleep efficiency, and sleep duration in older adults [9,22,23,24,25]. By contrast, moderate to vigorous intensity physical activity (MVPA) programs tend to positively affect sleep outcomes in older adults even more when compared with LPA. A recent review in older adults [9] showed that performing regular MVPA resulted in the highest number of significant improved sleep outcomes, such as quality of sleep, sleep latency, sleep disturbances, wake time after sleep onset, sleep duration, and sleep efficiency [26,27,28,29,30,31,32,33,34]. Despite the existing evidence on associations between PA and sleep, the associations of specific PA intensities and sleep remain unclear.

Besides PA, sedentary behaviour (SB) can also affect sleep. SB has been defined as any waking behaviour characterized by an energy expenditure ≤ 1.5 metabolic equivalents while in a sitting, reclining, or lying posture [34]. Contrary to PA, increased SB has been associated with worse sleep outcomes, such as insomnia, sleep disturbances [35,36,37,38], and insufficient sleep in older adults [37,38]. Given the associations of different intensities of PA and SB with sleep, there is a need to examine the associations between these behaviours and sleep at the same time. This is in line with the WHO’s healthy ageing recommendations that focus on reducing SB and increasing different intensities of PA for better health outcomes [9,19,39].

Therefore, in order to discover new insights in associations between movement behaviours and sleep, it is essential to examine time reallocation between different combinations of movement behaviours, such as SB, LPA, and MVPA. An isotemporal substitution model (ISM) can explore if, how, and to what extent changes in time spent in certain movement behaviours are associated with sleep outcomes in older adults. This novel statistical method allows investigation of the associations of absolute time reallocation (i.e., replacing time from one movement behaviour to another) between movement behaviours with health outcomes [39,40,41,42]. Grgic et al. (2018) reviewed and summarized a number of studies that examined associations of time reallocations with isotemporal substitution models [39]. This type of analysis was most often used to describe associations of time reallocations between SB, PA, and sleep with a wide range of health outcomes, such as biomarkers, pain, physical fitness, mortality, mental health, and quality of life [39,43,44,45,46].

Whilst the association between movement behaviours and sleep with health outcomes has been examined in children [45], adults [46], middle aged adults [43,44], and older adults [43,44,45,47], no study to our knowledge has yet examined the associations of reallocating time between SB and different intensities of PA with objective and subjective sleep outcomes in a large sample of older adults. This is a clear gap in the current literature. Such isotemporal analyses can help to fill this gap in the current knowledge, as they will aid in forecasting the optimal combination between time spent in either movement behaviour (such as SB, LPA, and MVPA) and sleep outcomes. Such improvement in our knowledge is important because of (1) the increasing number of older adults globally and the known positive associations between PA and sleep in older adults, (2) the public health burden of age-associated declines in sleep and the high prevalence of sleep problems, and (3) the far-reaching consequences of sleep problems for all aspects of health and the reliance on sleep medication, often accompanied by negative side effects in older adults.

Therefore, the aim of this study is to examine the associations of time replacement between device-based assessed SB and PA (of light and moderate to vigorous intensity) with objective and subjective sleep outcomes in older adults.

## 2. Materials and Methods

### 2.1. Design and Sample

Cross-sectional data were gathered from community-dwelling older adults (aged ≥ 55 years) from July 2018 to July 2019. These adults were recruited at weekly meetings held by a community service organization providing PA and socio-cultural activities for older adults in Flanders (OKRA SPORT+). Exclusion criteria were older adults who were not able to attend the meetings due to limited physical mobility. The ethics committee of UZ Leuven granted approval for this study (Ref. no: S61581). All participants received study information prior to the study and granted written informed consent.

### 2.2. Variables and Measurement

Data were collected using self-administered questionnaires (demographics and Pittsburg Sleep Quality Index (PSQI)) and accelerometers (sleep efficiency, wake after sleep onset (WASO), and number and length of awakenings).

#### 2.2.1. Questionnaires

Demographic variables (age, gender, education, and marital status), as well as general health information (smoking, use of alcohol, caffeine or screen time before bedtime, use of sleep medication, and presence of chronic conditions), were collected by means of a questionnaire. The definition of categories for education are in line with the International Standard Classification of Education (ISCED) 2011 [48]. Sleep quality was assessed with the PSQI questionnaire, a frequently used reliable and validated 19-item self-reported questionnaire in a variety of adult and older adult populations [48,49,50,51,52,53,54]. This questionnaire determines subjective sleep quality over the last month, and contains seven sleep characteristics, including sleep quality, sleep latency, sleep duration, habitual sleep efficiency, sleep disturbances, use of sleep medication, and daytime dysfunction. The total PSQI score ranges from 0 to 21 points, with five points or higher indicating poor-quality sleep [48,50]. Compared with other subjective measuring methods, the PSQI is easy to complete for older adults, and provides highly reliable and valid measures of sleep quality (Cronbach’s α 0.83) [52,53,54].

#### 2.2.2. Accelerometers

Movement behaviours (SB, LPA, and MVPA) and sleep outcomes (sleep efficiency, WASO, and amount and length of awakenings) were measured through accelerometry (Actigraph type wGT3X-BT, Actigraphcorp, Pensacola, FL, USA). The Actigraph wGT3X-BT is a wrist-worn accelerometer that measures and records physical movement associated with daily activity and sleep [55]. All participants were asked to wear the Actigraph device on their non-dominant wrist for six consecutive days, including two weekend days and five nights. The Actigraph wGT3X-BT has been used in numerous studies to measure SB, PA, and sleep in older adults, and resulted in valid measurements for this target population [56,57,58,59,60]. Accelerometer data were processed using well-established validated algorithms available in the Actilife software package (Actilife, v6.13.4) for wear time validation [56], PA [57,61] and sleep/wake identification [62]. Only data gathered over a minimum of four wear days of at least 10 h of waking wear time data were included in the analysis [56,62,63].

### 2.3. Analysis

Associations between different time use within specific types of movement behaviours and sleep outcomes in older adults were examined using isotemporal substitution models (ISMs). ISMs were first introduced in 2009 [40], and make it possible to examine associations of absolute time reallocation (i.e., 30 min) between movement behaviours (i.e., SB, LPA, and MVPA) with both objective and subjective sleep as outcomes. These reallocations of time are based on statistical modelling rather than on real-life changes in movement behaviours. More specifically, in this paper, we examined the associations between 30-min time reallocations from SB to LPA, from LPA to MVPA, and from SB and MVPA with subjective sleep quality and objectively measured sleep efficiency, WASO, and the number and length of awakenings. The behaviour in which time was being reallocated from was omitted from the model. Resulting coefficients of the remaining behaviours represented the association of reallocating 30 min from one behaviour (omitted) to another behaviour (included). These analyses are not indicative of individual changes in behaviour, rather, they model a theoretical shift in behaviour at a population level.

Three multiple linear regression models were composed: (1) a crude model (unadjusted without potential control variables), (2) a partially adjusted model adjusted for demographics, i.e., age, gender, education, and marital and professional status, and (3) a fully adjusted model adjusted for demographics and covariates that have shown to impact sleep, i.e., smoking [64,65,66], the use of alcohol [67,68,69,70], caffeine [71], screen time before bedtime [72,73], the use of sleep medication [74], and the presence of chronic conditions [75]. Our data met the assumptions that apply for linear regression models. All analyses were performed using SPSS version 24.0. Statistical significance was set at *p* ≤ 0.05.

## 3. Results

In total, 453 older adults participated in this study. Due to the minimum wear time compliance of four wear days of at least 10 h, data from 439 older adults (97%) were eligible for analysis. The study sample therefore consisted of 439 older adults (mean age: 71.6 years), of which 28% were males and 71% were females. See Table 1 for a detailed overview of participant characteristics. In short, 45% of the study sample was low-educated, and the majority of the sample were married or living together (76%) and were no longer professionally active (94%). Most of the study participants were non-smokers (94%) and did not use any alcohol (74%) or caffeine (82%) before bedtime. In 89% of the participants, screen time was present before bedtime. Although 53% of the participants reported the presence of chronic conditions, only 13% of the participants tended to take sleep medications. Movement behaviours and sleep parameters are reported in Table 2.

The results from all three ISM models (unadjusted, partially adjusted, and fully adjusted) are reported in Table 3. There were no significant associations between reallocations from SB to LPA, SB to MVPA, and LPA to MVPA with WASO, length of awakenings, and sleep quality (PSQI). For sleep efficiency, there was a significant negative association (i.e., lower efficiency) of replacement of 30 min of SB with LPA in the unadjusted (B = −6.19; 95% CI: −9.62, −2.77), partially adjusted (B = −5.89; 95% CI: −9.42, −2.37), and fully adjusted models (B = −5.36; 95% CI: −9.31, −1,40). Furthermore, there was a significant positive association (i.e., higher efficiency) of replacing 30 min of LPA with MVPA for sleep efficiency in the unadjusted model (B = 5.05; 95% CI: 1.53, 8.57). There also was a significant positive association (i.e., increased number of awakenings) of replacing 30 min of SB to MVPA with the number of awakenings in the unadjusted model (B = 0.25; 95% CI: 0.07, 0.43).

## 4. Discussion

This is the first large-scale study that used isotemporal substitution analysis to examine the associations of reallocating device-based measurement of SB and different intensities of PA with objective and subjective sleep in older adults. There were significant associations of replacing time between movement behaviours with sleep efficiency and the number of awakenings. More specifically, replacing 30 min of SB to LPA was associated with lower sleep efficiency in the unadjusted, the partially adjusted, and the fully adjusted models. By contrast, replacing 30 min of LPA with MVPA was associated with better sleep efficiency in the unadjusted model. Furthermore, replacing 30 min of SB with MVPA was associated with an increased number of awakenings in the unadjusted model. There were no statistically significant associations of replacing movement behaviours for wake time after sleep onset (WASO), length of awakenings, or sleep quality.

To our knowledge, the associations with sleep in older adults have been studied before in a smaller sample of Japanese adults [47]. This Japanese study included 70 adults aged 65+ years who did not have pre-existing diagnosed sleep problems and who did not use sleep medication. In that study, SB, PA, and sleep were also measured using Actigraph accelerometers and the PSQI. Their findings supported the positive association of replacing 30 min of SB with MVPA with the number of awakenings in our study. However, these authors reported additional associations of replacing 30 min of SB with LPA with better WASO, improved sleep fragmentation, and improved PSQI. Furthermore, replacing 30 min of SB with LPA was associated with an increased sleep efficiency in that study, whereas we found a negative association for sleep efficiency for this specific time reallocation. It should be noted though that we did not exclude older adults with pre-existing diagnosed sleep problems nor older adults who did use sleep medication. Moreover, compared with the participants in our study, participants in this Japanese study showed lower MVPA levels, a lower sleep efficiency, a higher WASO, and a higher number of awakenings. These differences in PA and sleep outcomes could account for the divergent results, because there was more room for improvement in the Japanese participants’ sleep.

The lack of more significant associations of time reallocations in this study may be surprising given the previously observed positive associations of both LPA and MVPA with WASO, sleep quality, sleep latency, and sleep disturbances [9,17,18,19], and given the negative associations of SB with sleep efficiency and sleep disturbances [35,36,37,38]. Potential explanations for this lack of more significant associations include (1) the measurement of SB, PA, and sleep, (2) the type of analysis (ISM), and (3) the characteristics of the included study sample in this study.

### 4.1. The Measurement of SB, PA, and Sleep

We collected subjective sleep data using a self-reported questionnaire (PSQI for sleep quality). Previous studies have already shown that the PSQI is a valid and reliable self-reported questionnaire to measure sleep quality in older adults [48,50,51,52,53,54]. It is well accepted by older adults and widely used. For example, a recent review summarizing the effects of PA programs on sleep in older adults showed that the PSQI was used as the main outcome measure in all but one study [9].

Furthermore, we collected SB, PA, and sleep data with accelerometers. Accelerometry is considered the standard for objectively measuring SB, and has been shown to provide valid estimates of SB. Wrist-worn Actigraph accelerometers have been widely used for measuring PA, and have been shown to increase wear compliance in participants in free-living conditions [76,77,78,79,80]. In terms of sleep, we realize that polysomnography is considered the golden standard for measuring sleep objectively, providing detailed information about different sleep stages and sleep patterns [81,82,83]. However, collecting polysomnographic data is time-consuming and expensive, and therefore not suitable for large-scale studies [81,82,83]. The use of (non-dominant) wrist-worn accelerometers provides sleep data when people reside in their own natural environment, and has also been shown to be valid in older adults [79,82,84]. Although, one may argue that the placement of the accelerometer on the wrist, rather than on the upper leg, may have affected the availability to distinguish SB from LPA, as the accuracy of measuring SB using accelerometers may depend on the wear location [85,86,87].

As stated in previous studies, objective and subjective sleep measures should ideally be combined to obtain comprehensive insight into different aspects of sleep quality and quantity. Interestingly, in the present paper, there were only statistically significant associations of reallocating time with device-based measured sleep efficiency and the number of awakenings. There were no statistically significant associations for the subjectively measured sleep quality, despite the reported associations between PA and sleep quality in previous research [9,47].

Despite the fact that the Actigraph wGT3X-BT has been shown to provide valid measurements for this target population for SB, PA, and sleep [55,56,57,58,59], using a single device that was wrist-worn could have affected the measurement of SB, PA, and sleep in this study, as the cut-off between SB and LPA and SB and sleep could be too closely aligned and difficult to distinguish [80,88]. This could have influenced the results in this study. Further research should therefore focus on integrating methods to assess SB, PA, and sleep throughout 24 h using a single device.

### 4.2. The Type of Analysis (ISM)

Isotemporal substitution analysis has been used in different populations with different health outcomes, such as mortality, general health status, mental health, adiposity, physical fitness, cardiometabolic health, and sleep [39,43,44,45,89,90]. According to a recent review in 2018 [39], the exchanged time between movement behaviours in studies using ISM varied from one minute to 120 min, with 30 min being the most common time reallocation. From a public health perspective, reallocating 30 min between movement behaviours seems more feasible and durable than longer periods of time (i.e., 60 min) to integrate in daily life situations, which is critical to sustain behaviour change in the long run [39,91,92]. Moreover, although vigorous intensity PA is also recommended for older adults on a weekly basis, we did not analyse the reallocation of time to vigorous intensity PA because of the low amounts of this type of intensity in our sample. This is a reflection of general population levels of PA [91,93,94], indicating that vigorous intensity activities are harder to maintain for longer periods of time and are therefore not durable in a daily lifestyle for older adults [91,95,96]. However, we reallocated time between SB and LPA with MVPA, which itself includes both moderate and vigorous intensity PA.

### 4.3. The Characteristics of the Study Sample

The demographic characteristics of the study sample are comparable to the Belgian population, as the majority of the population of older adults are also female (56%), married (56%), and no longer professionally active at the age of 65 years [97]. The average participant in this study was more physically active and less sedentary compared with the general Belgian population [91,93]. This could have affected the results in this study, as older adults with low PA levels are more likely to benefit from the reallocation of time, as this constitutes a larger proportional change. Future research should therefore use a random sampling procedure resulting in a more representative study sample in terms of PA levels. There should also be a specific effort to include the very old segment of the population (85+) given the rising prevalence of both physical inactivity and sleep problems with older age [1,2,98,99].

In terms of sleep, 13% of the participants used sleep medication on a daily basis in the present study. The average sleep efficiency of 94% was well above the cut-off for effective sleep of 85% [100]. Interestingly, the relatively low average PSQI global score for sleep quality of six points out of 21 indicates poor sleep [48,50,51,52]. Thus, our study sample showed efficient sleep but poor sleep quality. It should be noted again that we did not exclude participants with pre-existing sleep problems, nor did we assess this in the questionnaire. However, we did adjust our analysis and controlled for sleep medication by adding this as a covariate in the fully adjusted models. Previous research showed a high prevalence of sleep problems when people were physically inactive [98]. Therefore, including only older adults with pre-existing sleep problems could have resulted in more positive significant associations, as problematic sleepers could experience a larger margin benefit when increasing their PA levels.

### 4.4. Potential Mechanisms for an Association between SB, PA, and Sleep

Potential mechanisms for a beneficial link between PA and sleep include the promotion of relaxation, blood circulation, and energy expenditure. In turn, these changes are beneficial to initiating and maintaining sleep [9,20,21]. Research also showed positive links between SB and sleep; relaxing before bedtime is often done in a sitting, reclining, or lying position, and can be beneficial for sleep, as it facilitates relaxation and helps to slow down from the day [99,101,102,103,104]. Moreover, relaxing decreases stress hormones and allows the body to prepare for sleep [99,102,103,104]. SB prior to bedtime could therefore be beneficial for sleep outcomes. Interestingly, the vast majority of our study sample (89%) did use screen time as a form of SB prior to bedtime. Compared with the earlier mentioned positive links between SB and sleep, screen time is associated with worse sleep outcomes because of the blue light rays [36,71,72,105]. Therefore, depending on the specific timing and type of SB, there may be different associations with aspects of sleep. The Actigraph measurements in our study did not allow us to define the type of SB. This could be an interesting direction for further research given the importance of the quality and timing of SB and its effect on sleep. Furthermore, the strength of association between PA and sleep differs between specific sleep outcomes, with WASO, sleep quality, sleep latency, and sleep disturbance showing the highest proportion of significant improvement after PA [9].

### 4.5. Strengths and Limitations

This study has several strengths and limitations. Strengths include the large sample size, the availability of subjective and objective sleep data, and the application of a novel statistical approach ISM to examine replacement effects.

Limitations include generalizability, the cross-sectional design, potential for type 1 error, and the use of one device that is wrist-worn. First, the study sample was recruited from one single socio-cultural organization that showed to be more physically active compared with the general population of older Belgian adults. Second, SB, PA, and sleep outcomes were measured at the same time in this cross-sectional design. Although we examined the associations of movement behaviours with sleep outcomes, we cannot exclude the possibility of bi-directional associations. Although SB and PA that were performed before bedtime could have affected sleep outcomes, we were not able to control for the exact timing of SB and PA in this analysis. Third, although we used a comprehensive set of objective and subjective sleep outcomes, we performed several tests in different models. Therefore, we cannot exclude the possibility of a type 1 error, assuming that, based upon a set α 0.05, there might be a 5% possibility that significant findings might be based on chance rather than on significance. The fact that the associations with sleep efficiency and the number of awakenings are also found in a previous study support real associations rather than type 1 errors. Fourth, we used one wrist-worn device (Actigraph) to measure SB, PA, and sleep outcomes. Despite the fact that there were no collinearity issues between movement behaviours and sleep data in this study, using wrist-worn devices to measure SB does not allow us to define the exact type of SB (i.e., sitting, reclining, or lying). If we had been able to collect information about the type of SB and examine reallocations between different types of SB, according to their sleep promoting characteristics with different intensities of PA, we could have found more specific associations for each type of SB.

### 4.6. Generalizability and Implications

The conclusions from this study apply to generally healthy older adults rather than to older adults with specific sleep problems or chronic conditions. Moreover, it should also be taken into account that our analyses were based on statistical modelling rather than on real-life changes in movement behaviours.

## 5. Conclusions

This study showed associations of time replacement by using isotemporal substitution analysis: (1) of replacing 30 min of LPA with MVPA with improved sleep efficiency; (2) of replacing 30 min from SB to MVPA with an increased number of awakenings; and (3) of replacing 30 min from SB to LPA with decreased sleep efficiency. There were no significant associations of time reallocations for WASO, length of awakenings, and sleep quality. Although it should be emphasized that we examined associations of modelled time reallocations with sleep, the results from this study improve our understanding of the interrelationships between different movement behaviours and sleep in older adults.

## Figures and Tables

**Table 1 ijerph-17-09579-t001:** Characteristics of participants.

Characteristics	Characteristics of Participants (n = 439) *	
Age, mean (SD) Sample range		71.6 (± 6.5) 55.7–94.1
Sex, n(%)	Male Female	125 (28%) 313 (71%)
Education*, n(%)	Low Medium High	196 (45%) 140 (32%) 85 (19%)
Marital status, n(%)	Married/cohabitant Single	334 (76%) 102 (23%)
Professionally active, n(%)	Yes No	19 (4%) 414 (94%)
Smoking, n(%)	Yes No	14 (3%) 414 (94%)
Alcohol before bedtime, n(%)	Yes No	99 (23%) 326 (74%)
Caffeine before bedtime, n(%)	Yes No	63 (14.5%) 361 (82%)
Screen time before bedtime, n(%)	Yes No	390 (89%) 28 (6%)
Sleep medication, n(%)	Yes No	57 (13%) 372 (85%)
Chronic condition, n(%)	Yes No Missing	169 (38%) 232 (53%) 38 (9%)

* Only percentages of missing values > 5% were separately reported. Column percentages for the separate variables may therefore not add up to 100%.

**Table 2 ijerph-17-09579-t002:** Movement behaviours and sleep parameters.

Parameter	Mean (SD)
**Movement Behaviours**	
SB (minutes/day)	603.16 (± 82,77)
LPA (minutes/day)	392.89 (± 61,42)
MVPA (minutes/day)	190.16 (± 75,83)
**Objective sleep parameters**	
Sleep efficiency (%)	93.84 (± 2.93)
WASO (minutes)	22.42 (± 11.31)
Number of awakenings	9.74 (± 4.70)
Length of awakenings (minutes)	22.42 (± 11.31)
**Subjective sleep parameters**	
PSQI global score, point *	6 (± 4)

* The PSQI global score ranges from 0 to 21 points, with five points or higher indicating poor-quality sleep [52].

**Table 3 ijerph-17-09579-t003:** ISM models for 30 min time replacement between SB, LPA and MVPA; and sleep outcomes.

Sleep Outcome	SB to LPA		SB to MVPA		LPA to MVPA	
	B	(95% CI)	B	(95% CI)	B	(95% CI)
**Sleep efficiency (in %) (unadjusted)**	**−6.19**	**(−9.62; −2.77) ***	−0.63	(−3.39; 2.14)	**5.05**	**(1.53; 8.57) ***
Partially adjusted	**−5.89**	**(−9.42; −2.37) ***	−2.36	(−5.32; 0.61)	3.03	(−0.75; 6.81)
Fully adjusted	**−5.36**	**(−9.31; −1.40) ***	−1.76	(−4.97; 1.44)	2.74	(−1.44; 6.92)
**Wake time after sleep onset (WASO) (in minutes) (unadjusted)**	−0.17	(−0.71; 0.36)	0.19	(−0.24; 0.62)	0.35	(−0.20; 0.90)
Partially adjusted	−0.13	(−0.66; 0.41)	−0.19	(−0.64; 0.26)	−0.12	(−0.69; 0.45)
Fully adjusted	−0.02	(−0.63; 0.58)	−0.03	(−0.52; 0.46)	−0.14	(−0.78; 0.50)
**Number of awakenings (unadjusted)**	0.07	(−0.15; 0.30)	**0.25**	**(0.07; 0.43) ***	0.21	(−0.01; 0.44)
Partially adjusted	0.06	(−0.16; 0.27)	0.08	(−0.10; 0.26)	0.04	(−0.20; 0.27)
Fully adjusted	0.08	(−0.16; 0.33)	0.12	(−0.08; 0.31)	0.01	(−0.25; 0.26)
**Length of awakenings (in minutes) (unadjusted)**	−0.17	(−0.71; 0.36)	0.19	(−0.24; 0.62)	0.35	(−0.20; 0.90)
Partially adjusted	−0.13	(−0.66; 0.41)	−0.19	(−0.64; 0.26)	−0.12	(−0.69; 0.45)
Fully adjusted	−0.02	(−0.63; 0.59)	−0.03	(−0.52; 0.46)	−0.14	(−0.78; 0.50)
**Sleep quality PSQI (unadjusted)**	−0.00	(−0.18; 0.17)	−0.10	(−0.23; 0.05)	−0.12	(−0.30; 0.06)
Partially adjusted	0.03	(−0.15; 0.20)	−0.11	(−0.26; 0.04)	−0.16	(−0.35; 0.03)
Fully adjusted	0.06	(−0.13; 0.25)	−0.05	(−0.20; 0.10)	−0.15	(−0.35; 0.04)

Model 1: unadjusted; Model 2: partially adjusted (demographics); Model 3: fully adjusted (demographics and covariates for sleep); B = beta; * significant at α ≤ 0.05 and are indicated in bold.

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
