# Peer review of "Are Reallocations between Sedentary Behaviour and Physical Activity Associated with Better Sleep in Adults Aged 55+ Years? An Isotemporal Substitution Analysis"

_ijerph, 2020, doi:10.3390/ijerph17249579_

Round 1
Reviewer 1 Report
“This is a cross-sectional, observational study whose aim is to examine the associations of time-replacement between device-based assessed SB and PA (of light and moderate-to-vigorous intensity) with objective and subjective sleep outcomes in older adults (aged ≥55years).” Results were “no association” for the majority of associations examined.
It was a pleasure to read such a well-written paper. Tables and text were easy to follow.
Now for the inevitable criticisms!
METHODS
Section 2.2.2. It would be helpful to other researchers if you could comment briefly on the level of compliance among your participants. You had data from 439 who managed to complete four 10-hr days of accelerometer wearing, but how many did you try to enroll who did not complete the regimen?
The description of the methods could be improved: the phrase “replacing 30 minutes of SB with 30 minutes of __” makes the reader think that there is an active substitution going on, as in an intervention study where participants are told to actively substitute a certain level of PA for a normally sedentary 30-minute period of their day. As I understand your methods, though, this study is completely observational and cross-sectional (as finally clarified in section 4.6). Perhaps those more familiar with ISM would understand this, but I found the wording somewhat misleading.
Another thing is unclear: In your analyses, did you first create scores for each individual of their mean number of minutes per day of SB, LPA, MVPA and their personal sleep outcomes, and then use those in your regressions, or did you allow multiple (e.g., five days of) PA observations per person and examine within-person effects as well? If the latter, did you cluster the observations by ID?
DISCUSSION
Section 4.5 Nice coverage of strengths and weaknesses.
Related to that, what about the time of day of the PA relative to bedtime? Could that have an effect on sleep that should be controlled for as an interaction term in your analyses? You mentioned that type of SB (e.g., screen time) right before bed might have an effect, but I know from my own experience that a vigorous workout right before bed will delay sleep. There must be some literature on this subject. Perhaps this is a topic you could examine in a future paper with the data you have already collected. In any case, I think it should be mentioned here as a limitation if not examined.
A few suggestions for language clarity:
54-55 Replace “Light intensity physical intensity” with “Light intensity physical activity”
71 Replace “focuses” with “focus”
77 Replace “to investigate” with “investigation of”
80 Replace “allow to help forecast” with “aid in forecasting”
123 Replace “high” with “highly”
130 Replace “of which two weekend days” with “including two weekend days”
Multiple lines: Replace “chronical conditions” with “chronic conditions” (unless this is a Briticism?)
219 Replace “has shown” with “has been shown”
220 Replace “The use wrist-worn Actigraph for measuring PA has been widely used and has shown” with “Wrist-worn Actigraph accelerometers have been widely used for measuring PA and have been shown”
226 Replace “has shown” with “has been shown”
236 Replace “has shown” with “has been shown”
Author Response
Reviewer #1
It was a pleasure to read such a well-written paper. Tables and text were easy to follow.
We thank Reviewer 1 for these positive comments. They are much appreciated.
Now for the inevitable criticisms!
We thank reviewer #1 for these constructive comments. In the points below we give a detailed overview of the answered or argumented comments.
METHODS
- Section 2.2.2. It would be helpful to other researchers if you could comment briefly on the level of compliance among your participants. You had data from 439 who managed to complete four 10-hr days of accelerometer wearing, but how many did you try to enroll who did not complete the regimen?
We agree that adding information on wear time compliance could be helpful to other researchers and readers. We have therefore added this information in lines 178-179: “In total, 453 older adults participated in this study. Due to the minimum wear time compliance of four wear days of at least 10 hours, data from 439 older adults (97%) were eligible for analysis.”
- The description of the methods could be improved: the phrase “replacing 30 minutes of SB with 30 minutes of __” makes the reader think that there is an active substitution going on, as in an intervention study where participants are told to actively substitute a certain level of PA for a normally sedentary 30-minute period of their day. As I understand your methods, though, this study is completely observational and cross-sectional (as finally clarified in section 4.6). Perhaps those more familiar with ISM would understand this, but I found the wording somewhat misleading.
The isotemporal analysis allows to examine the associations of reallocations of time between SB and PA, however, this analysis is statistical and is not based on real-life time reallocations, as was already stated in the discussion section (line 353). In order to help the reader understand this analysis, we now also clarified this in the abstract in lines 16-18: “This study examines the associations of statistically modelled time reallocations between sedentary time and different physical activity intensities with sleep outcomes using isotemporal substitution analysis.”
Furthermore, we also added this in the methods section in lines 160-161 as follows: “The reallocations of time are based on statistical modelling, rather than on real-life changes in movement behaviours.”
- Another thing is unclear: In your analyses, did you first create scores for each individual of their mean number of minutes per day of SB, LPA, MVPA and their personal sleep outcomes, and then use those in your regressions, or did you allow multiple (e.g., five days of) PA observations per person and examine within-person effects as well? If the latter, did you cluster the observations by ID?
Thank you for pointing out this ambiguity We did not allow multiple observations per participant and only used mean values (minutes) per SB, LPA and MVPA and participants’ sleep outcomes in the regression analysis. Hence, we believe that cluster analysis by ID is redundant. Moreover, our analyses are not indicative of individual changes in behaviour, rather they model a theoretically shift in behaviour at a population level. In response to this comment, we clarified this in the methods section in lines 167-168 as follows: “These analyses are not indicative of individual changes in behaviour, rather they model a theoretically shift in behaviour at a population level.
DISCUSSION
- Section 4.5 Nice coverage of strengths and weaknesses.
Related to that, what about the time of day of the PA relative to bedtime? Could that have an effect on sleep that should be controlled for as an interaction term in your analyses? You mentioned that type of SB (e.g., screen time) right before bed might have an effect, but I know from my own experience that a vigorous workout right before bed will delay sleep. There must be some literature on this subject. Perhaps this is a topic you could examine in a future paper with the data you have already collected. In any case, I think it should be mentioned here as a limitation if not examined.
Thank you for bringing this to our attention. We learned from prior studies that vigorous physical activity right before bedtime could indeed (negatively) affect sleep. In our study, the proportion of participants engaging in vigorous intensity activity was very low, namely <1%. Furthermore, we examined mean values of different PA intensities, rather than the patterns and timing of physical activity at different intensities. Although beyond the scope of this study, this could indeed be an interesting avenue for future research. Therefore, we now mention this in the limitations of the revised manuscript in lines 336-338: ”Although SB and PA that was performed before bedtime could have affected sleep outcomes, we were not able to control for the exact timing of SB and PA in this analysis.”
- A few suggestions for language clarity:
Thank you for bringing these corrections to our attention. All typographical changes have been corrected in the revised text according to the suggestions below:
- 60 Replace “Light intensity physical intensity” with “Light intensity physical activity”
- 76 Replace “focuses” with “focus”
- 82 Replace “to investigate” with “investigation of”
- 95 Replace “allow to help forecast” with “aid in forecasting”
- 140 Replace “high” with “highly”
- 148 Replace “of which two weekend days” with “including two weekend days”
- Multiple lines: Replace “chronical conditions” with “chronic conditions” (unless this is a Briticism?) (lines 174,186, Table 1)
- 248 Replace “has shown” with “has been shown”
- 249-251 Replace “The use wrist-worn Actigraph for measuring PA has been widely used and has shown” with “Wrist-worn Actigraph accelerometers have been widely used for measuring PA and have been shown”
- 250 Replace “has shown” with “has been shown”
- 256 Replace “has shown” with “has been shown”
Reviewer 2 Report
This study examines the associations of time reallocations between sedentary time and different physical activity intensities with sleep outcomes using isotemporal substitution analysis. Device-measured physical activity data and both objective and subjective sleep data were collected from 439 adults ages 55+ years. Replacing 30 minutes of sedentary behaviour with moderate to vigorous intensity physical activity (MVPA) was significantly associated with an increased number of awakenings. Moreover, a reallocation of 30 minutes between light physical activity (LPA) and MVPA was significantly associated with increased sleep efficiency. Furthermore, reallocating 30 minutes of sedentary behaviour to LPA showed a significant association with decreased sleep efficiency. There were no significant associations of time reallocations for wake time after sleep onset, length of awakenings and sleep quality.
This is an interesting study with very promising contribution to the field. The sample is large and I appreciate the rigor of the analyses that were conducted. The interpretation and discussion of the result was also carefully done. I agree that the paper is suitable for publication.
One major comment is related to the use of isotempral substitution analysis. I feel that it will be good for the readers if the authors can elaborate further how the analysis was conducted and what are the assumptions of the analysis.
I only have a few minor comments:
pp 1, line 35. Napping in itself isn’t exactly a problem – authors could specify in what context would napping be considered problematic
pp 2, line 44. Definition of PA should be delineated here.
pp 2, lines 81 – 84. Another importance of the improvement in knowledge should be less reliance on medication due to their side effects as delineated in the last lines of the 1st paragraph.
pp 2, lines 89 – 93. This section can be shifted up to the previous paragraph to improve the logical flow and avoid repetition of certain points, such as “the clear gap in the current literature due to the lack of such studies”, “rising prevalence of sleep problems in older adults”, so on.
pp 3, lines 996-98. It will be good to elaborate the hypotheses/aims further.
pp 7, line 220. “The use wrist-worn Actigraph...” – “use” to be omitted.
pp 8, line 282. Sentence can be re-constructed such that it flows logically – “Research also showed (positive) links between SB and sleep: relaxing before bedtime…”
Author Response
Reviewer #2
This is an interesting study with very promising contribution to the field. The sample is large and I appreciate the rigor of the analyses that were conducted. The interpretation and discussion of the result was also carefully done. I agree that the paper is suitable for publication.
We would like to thank Reviewer #2 for the interest in our article, this positive feedback and the given remarks and suggestions to further improve the manuscript.
- One major comment is related to the use of isotemporal substitution analysis. I feel that it will be good for the readers if the authors can elaborate further how the analysis was conducted and what are the assumptions of the analysis.
Thank you for bringing this to our attention. We have now added information on what exactly was done when conducting the analysis in lines 164-167: “The behaviour in which time was being reallocated from, was omitted from the model. Resulting coefficients of the remaining behaviours represent the association of reallocating 30 minutes from one behaviour (omitted) to another behaviour (included).”
In terms of assumptions, prior to the analysis we checked whether our data met the assumptions for linear regression. We have now included this information in the Methods section in lines 174-175 as follows: “Our data met the assumptions that apply for Linear Regression models.”
I only have a few minor comments:
- pp 1, line 35. Napping in itself isn’t exactly a problem – authors could specify in what context would napping be considered problematic
Napping becomes problematic when it takes up excess time during the day, thereby in turn influencing sleep at night. We have therefore added information and an additional reference on napping as follows in lines 35-36: “These changes in sleep are problematic as they cause fatigue, daytime sleepiness and excess napping during the day, which in turn can influence the sleep at night [6,7].”
- pp 2, line 44. Definition of PA should be delineated here.
We have delineated the definition of PA as suggested in line 45 as follows: “Physical activity (PA) has been defined by Caspersen et al.(1985) as any bodily movement produced by skeletal muscles that results in energy expenditure [14]. These movements can be classified as light, moderate, or vigorous intensity, depending on the amount of energy expended by an individual during the activity [14,15,16].”
- pp 2, lines 81 – 84. Another importance of the improvement in knowledge should be less reliance on medication due to their side effects as delineated in the last lines of the 1st paragraph.
We have added the aspect of reliance on sleep medication and their side effects in lines 96-101:
“Such improvement in our knowledge is important because of; 1) the increasing number of older adults globally and the known positive associations between PA and sleep in older adults; 2) the public health burden of age-associated declines in sleep and the high prevalence of sleep problems; and 3) the far-reaching consequences of sleep problems for all aspects of health and the reliance on sleep medication, often accompanied by negative side effects, in older adults.”
- pp 2, lines 89 – 93. This section can be shifted up to the previous paragraph to improve the logical flow and avoid repetition of certain points, such as “the clear gap in the current literature due to the lack of such studies”, “rising prevalence of sleep problems in older adults”, so on.
Thank you for pointing this out. We have improved the flow and now avoid repetition from 84-115 as follows: “Grgic et al. (2018) reviewed and summarized a number of studies that examined associations of time reallocations with isotemporal substitution models [39]. This type of analysis was most often used to describe associations of time reallocations between SB, PA and sleep with a wide range of health outcomes, such as biomarkers, pain, physical fitness, mortality, mental health and quality of life [39,43,44,45,46].
Whilst the association between movement behaviours and sleep with health outcomes has been examined in children [45], adults [46], middle aged [43,44] and older adults [43,44,45], no study to our knowledge, has yet examined the associations of reallocating time between SB and different intensities of PA with objective and subjective sleep outcomes in a large sample of older adults. This is a clear gap in the current literature. Such isotemporal analyses can help to fill this gap in the current knowledge as they will aid in forecasting the optimal combination between time spent in either movement behaviour (such as SB, LPA and MVPA) and sleep outcomes. Such improvement in our knowledge is important because of; 1) the increasing number of older adults globally and the known positive associations between PA and sleep in older adults; 2) the public health burden of age-associated declines in sleep and the high prevalence of sleep problems; and 3) the far-reaching consequences of sleep problems for all aspects of health and the reliance on sleep medication, often accompanied by negative side effects, in older adults.
Therefore, the aim of this study is to examine the associations of time-replacement between device-based assessed SB and PA (of light and moderate-to-vigorous intensity) with objective and subjective sleep outcomes in older adults.
- pp 3, lines 996-98. It will be good to elaborate the hypotheses/aims further.
Thank you for this comment. However, as this is the first study to examine associations of time reallocations between SB, PA and sleep in a large sample of older adults, our intention was to explore possible associations. We therefore did not propose any hypotheses prior to the analyses, as this would give direction to our findings.
- pp 7, line 220. “The use wrist-worn Actigraph...” – “use” to be omitted.
We have changed this sentence according to this suggestion in lines 249-251: “Wrist-worn Actigraph accelerometers have been widely used for measuring PA and have been shown to increase wear compliance in participants in free-living conditions [83,84,85].”
- pp 8, line 282. Sentence can be re-constructed such that it flows logically – “Research also showed (positive) links between SB and sleep: relaxing before bedtime…”
We have reconstructed this sentence in lines 312-313 as follows: “Research also showed positive links between SB and sleep: relaxing before bedtime is often done in a sitting, reclining or lying position and can be beneficial for sleep as it facilitates relaxation and helps to slow down from the day [101,103,104,105,106].”